# A comparison of biomonitoring methodologies for surf zone fish communities

**Zachary Gold**[1][ID][☯]*, **McKenzie Q. Koch**[1][☯], **Nicholas K. Schooler**[2], **Kyle A. Emery**[2], **Jenifer E. Dugan**[2], **Robert J. Miller**[2], **Henry M. Page**[2], **Donna M. Schroeder**[3], **David M. Hubbard**[2], **Jessica R. Madden**[2], **Stephen G. Whitaker**[2,4], **Paul H. Barber**[1]

1 Department of Ecology and Evolutionary Biology, University of California, Los Angeles, Los Angeles, CA, United States of America, 2 Marine Science Institute, University of California, Santa Barbara, Santa Barbara, CA, United States of America, 3 Bureau of Ocean Energy Management, Camarillo, CA, United States of America, 4 Channel Islands National Park, Ventura, CA, United States of America

☯ These authors contributed equally to this work.

* zjgold@ucla.edu

**Data Availability Statement:** All data, accession numbers, and code used to conduct analyses are publicly available on Dryad (https://doi.org/10.5068/D1SQ4Z), NCBI SRA (BioProject

## Abstract

Surf zones are highly dynamic marine ecosystems that are subject to increasing anthropogenic and climatic pressures, posing multiple challenges for biomonitoring. Traditional methods such as seines and hook and line surveys are often labor intensive, taxonomically biased, and can be physically hazardous. Emerging techniques, such as baited remote underwater video (BRUV) and environmental DNA (eDNA) are promising nondestructive tools for assessing marine biodiversity in surf zones of sandy beaches. Here we compare the relative performance of beach seines, BRUV, and eDNA in characterizing community composition of bony (teleost) and cartilaginous (elasmobranch) fishes of surf zones at 18 open coast sandy beaches in southern California. Seine and BRUV surveys captured overlapping, but distinct fish communities with 50% (18/36) of detected species shared. BRUV surveys more frequently detected larger species (e.g. sharks and rays) while seines more frequently detected one of the most abundant species, barred surfperch (*Amphistichus argenteus*). In contrast, eDNA metabarcoding captured 88.9% (32/36) of all fishes observed in seine and BRUV surveys plus 57 additional species, including 15 that frequent surf zone habitats. On average, eDNA detected over 5 times more species than BRUVs and 8 times more species than seine surveys at a given site. eDNA approaches also showed significantly higher sensitivity than seine and BRUV methods and more consistently detected 31 of the 32 (96.9%) jointly observed species across beaches. The four species detected by BRUV/seines, but not eDNA were only resolved at higher taxonomic ranks (e.g. Embiotocidae surfperches and Sygnathidae pipefishes). In frequent co-detection of species between methods limited comparisons of richness and abundance estimates, highlighting the challenge of comparing biomonitoring approaches. Despite potential for improvement, results overall demonstrate that eDNA can provide a cost-effective tool for long-term surf zone monitoring that complements data from seine and BRUV surveys, allowing more comprehensive surveys of vertebrate diversity in surf zone habitats.

PRJNA966946), and GitHub (https://doi.org/10.5281/zenodo.7888385).

**Funding:** Jennifer E. Dugan and Robert J. Miller received support from Bureau of Ocean Energy Management Environmental Studies Program (M15AC00012 and MC15AC00006) and NASA Biodiversity and Ecological Forecasting Program (NNX14AR62A). Jennifer E. Dugan also received support from National Oceanic and Atmospheric Administration. Robert J. Miller also received funding from the National Marine Environmental Monitoring Center, State Oceanic Administration. Paul H. Barber received support from the Howard Hughes Medical Institute. McKenzie Q. Koch received support from the University of California CALeDNA program and Holmes O. Miller Endowment Fund.

**Competing interests:** The authors have declared that no competing interests exist.

## Introduction

Sandy beaches and their adjacent surf zones comprise ~30% of the world's ice-free shoreline [1, 2]. Surf zones provide critical ecosystem services, supporting local marine biodiversity through the provisioning of forage habitat, refuge from predators, spawning sites, and nurseries for commercially and recreationally important fish species [1, 3–5]. Furthermore, sandy beaches and surf zones are important areas for recreation and tourism [4, 6, 7]. In California alone, the value of sandy beach ecosystem services in 2017 was estimated at $25.9 billion annually [7–9].

Despite their tremendous societal and ecological value, our understanding of the status, and spatial and temporal dynamics of surf zone fish communities in southern California and around the world is lacking [2], and sandy beaches and surf zones are rarely included in conservation management plans [10]. Sandy beaches and associated surf zone biological communities face both local and global anthropogenic stressors that threaten their biodiversity and ecosystem function [11]. Sea-level rise coupled with coastal armoring is contributing to coastal squeeze, compressing or eliminating sandy beaches and altering surf zone habitats [12–15]. Coastal urban development and engineering are increasing erosion along shorelines, increasing turbidity and altering surf zone characteristics [16–18]. Compounding these stressors, pollutants from stormwater, sewage, oil spills, and agricultural runoff often spill directly into surf zone habitats [11]. As urban development and climate change continues to impact these important coastal ecosystems, our ability to effectively manage sandy beaches hinges on accurate assessments and monitoring of the species and communities that depend on them [4, 11].

Traditional methods for monitoring surf zone ecosystems are based on surveys using nets, such as seines or bottom trawls, or hook and line fishing to capture surf zone fish [2, 4, 19]. Net, and hook and line surveys are advantageous as they can provide detailed information on size, sex, and age structure of fish populations, and are not influenced by poor underwater visibility. However, these capture surveys have known biases that limit their reliability and repeatability. Hook and line fishing surveys are often species-specific due to the choice of tackle and bait, and observer skill affects capture rates [20]. Wave and weather conditions can affect seine surveys by reducing the capture efficiency of nets and creating hazards to researchers in heavy surf (Table 1). Seines are also sensitive to slight variation in mesh size, width of opening, and speed of implementation, impacting repeatability and comparability of results [10, 21]. Seines are also less effective for sampling large, fast-moving species [22, 23] as well as small benthic fishes, such as flatfish (Families Pleuronectidae and Paralichthyidae), that pass through or under the nets. In addition, both these techniques are highly labor-intensive, and can be destructive, often injuring or killing captured specimens [24] (Table 1).

Alternative surf zone biomonitoring approaches rely on visual surveys, either via SCUBA or snorkel transects or baited remote underwater video (BRUV) units [2, 25, 26]. BRUVs are increasingly used to overcome diver avoidance behavior [19, 27–29], instead employing baited video cameras that record fish passing through the field of view, allowing for non-invasive measurements of fish diversity, abundance, and behavior. However, BRUV surveys also have biases that limit their reliability and repeatability (Table 1). Large waves, inclement weather, light conditions, and drifting macrophytes, can all reduce visibility and impair species identification and detection [30, 31]. BRUV methods are also sensitive to bait choice, length and location of deployment [10, 21], may not attract planktotrophic and herbivorous fish that are not attracted to the bait, and are poor at detecting cryptic species [21]. Moreover, they are challenging to deploy by kayak or swimming in the surf zone, and can require processing of hundreds of hours of underwater video [26]. Together, these limitations affect the reliably and effectiveness of visual monitoring approaches of surf zone fish communities, highlighting the need for new approaches.

**Table 1. Comparisons of survey methods.**

| Metric | Beach Seine | BRUV | eDNA |
|---|---|---|---|
| Team size needed | 4–6 | 2 | 2 |
| Set up and Field time | 20 minutes per seine, 20–85 minutes to measure & release | 1.5 to 2.0 hours | 20 minutes |
| Field Gear required | Seine, poles, lines | Weighted video rigs, bait | Sampling bags, filters, ice chest |
| Field Sample processing | Minimal, gear clean up and repair | Minimal, gear clean up and repair | ~1.5 hours for gravity filtering and preserving samples |
| Post-Field Sample Processing | None | 1.5–3.0 hours per video | Theoretical Minimum 24 hours per sample (DNA extraction, PCR, Library preparation, sequencing, and bioinformatics), but can be automated and optimized for high throughput |
| Sample Archiving | No–fish released | Yes–video record | Yes–DNA extractions archived & sequence record |
| Abundance | Relative | Relative | Relative (needs ground truthing) |
| Size and age distribution | Yes | No | No |
| Injury/mortality of fish | A small percentage of catch | No | No |
| Effect of sea conditions | Significant- affects net behavior and safety | Significant- affects visibility | Wider tolerance but unknown effects on spatial and temporal variability |

A promising new approach for surveying the diversity of coastal marine ecosystems is environmental DNA (eDNA) metabarcoding [32, 33]. eDNA refers to the collection, capture, sequencing, and identification of DNA shed from organisms inhabiting a particular ecosystem [34, 35]. Studies indicate that eDNA metabarcoding is highly sensitive and provides an accurate, practical, and cost-effective method of monitoring marine biodiversity [36–40].

Studies of eDNA highlight some key advantages relative to seining and BRUVs (Table 1). In particular, eDNA identifies a broad diversity of marine life, frequently detecting more species than other methods [41–44], including cryptic, rare, invasive, and endangered species [45–49], and is effective across a variety of marine ecosystems, including coral reefs [50, 51], kelp forests [52, 53], estuaries [31, 54, 55], and coastal oceans [39, 56, 57]. eDNA is largely independent of developmental stage, allowing for the detection of larval and juvenile life stages, identifying potential nursery grounds [32]. In addition, eDNA samples are simple to collect, encouraging student led and community science, and are also cost effective, permitting increased sampling efforts across both time and space [32, 58–60].

Yet eDNA also has limitations. For example, the need for molecular expertise and laboratory space to process samples may limit some research groups and monitoring agencies where such resources are not already available [36]. Additionally, eDNA does not provide key information needed for fishery and stock assessments (e.g., size, age, sex), and it is unclear whether eDNA results accurately reflect the relative abundance of marine species [31, 40, 61, 62].

There are also unresolved questions about the fate and transport of eDNA, particularly in highly dynamic coastal marine ecosystems. For example, previous studies report spatial resolution of eDNA in nearshore marine environments is on the scale of 50–1000 m [52, 53, 63–69] and temporal resolution is on the scale of hours to days [67, 70, 71], complicating the ecological interpretation of detected community assemblages [72]. However, these studies were not conducted in surf zone ecosystems which are strongly affected by wave driven longshore transport and nearshore currents with higher velocities (e.g., rip currents) and tides compared to the subtidal ecosystems previously studied, potentially integrating ecological signatures over greater space and time, and mixing species detections across ecosystems [2].

Although eDNA and BRUV surveys hold promise for monitoring surf zone habitats, evaluating how well these methods perform compared to traditional seine surveys and each other is a crucial information gap [26, 73, 74]; to date, only two studies [52, 75] employed eDNA to assess fish biodiversity in surf zones habitats. To address this gap, we compared the ability of seine, BRUV, and eDNA methods to describe surf zone fish communities using a series of surveys where we simultaneously employed all three methodologies at 18 open coast surf zones associated with beaches in southern California. We compared these results to assess how the different survey methods performed in surf zone habitats, information critical to resource managers charged with monitoring these important coastal ecosystems.

## Methods

### Study sites

To compare the effectiveness of seine, BRUV, and eDNA survey techniques for monitoring surf zone bony (teleost) and cartilaginous (elasmobranch) fish communities, we deployed the three survey techniques contemporaneously at 18 sandy beach sites across southern California, USA (Fig 1; S1 Table in S2 File); 14 on the California Channel Islands and 4 on the mainland. These represent novel fish community surveys for all but three of the mainland sites, providing important baseline data for fish assemblages. To maximize comparability, we surveyed surf

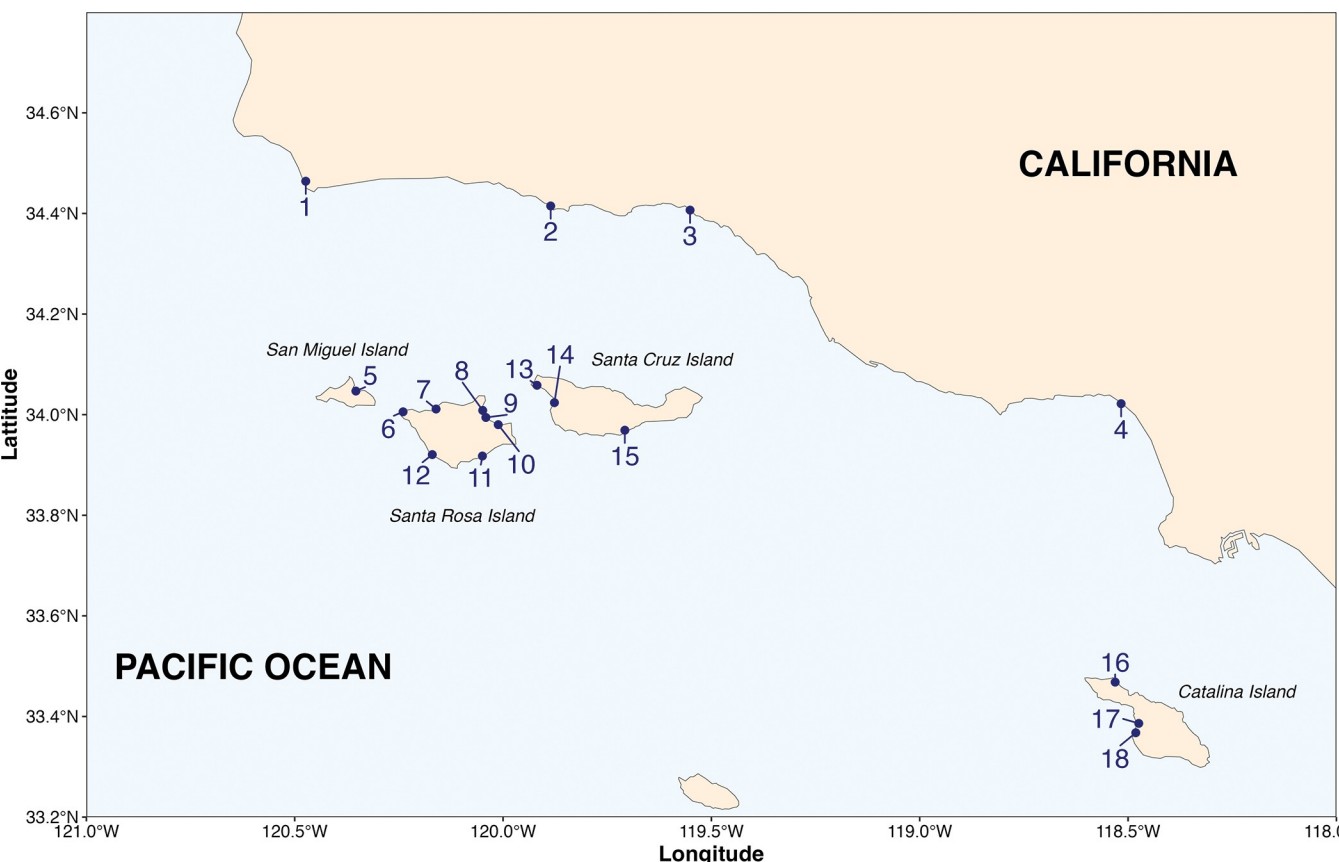

**Fig 1. Site map.** Map of the study region showing mainland sites, Northern Channel Islands sites, and Catalina Island sites on the coast of southern California, USA. Black dots and numbers correspond to site names. 1– Dangermond, 2 –R Beach, 3 –Santa Claus, 4 –Santa Monica, 5 –Cuyler Harbor, 6 –Sandy Point, 7 – Soledad, 8 –Bechers Bay, 9 –Water Canyon, 10 –Southeast Anchorage, 11 –Ford Point, 12 –China Camp, 13 –Forney Cove, 14 –Christy Beach, 15 –Coches Prietos, 16 –Emerald Bay, 17 –Little Harbor, 18 –Ben Weston. The base map was created with Natural Earth Dataset (http://www.naturalearthdata.com/).

zones using all three methods at each location on the same day using the methods described below. All surveys were conducted between August 15, 2018 and November 2, 2018. At one site, Soledad beach, on Santa Rosa Island, we were unable to conduct the BRUV surveys due to hazardous surf conditions.

## Beach seine surveys

Beach seine surveys were employed using methods modified from the California Department of Fish and Wildlife (Monterey, CA, USA) (Carlisle, Schott & Abrahamson, 1960) using a 15.3 m (50 ft) x 1.8 m (6 ft) seine net (10 mm knotless nylon mesh, 2 m poles, 20 m leader ropes) with a bag, floats, and weighted lead line. At each site, we conducted seine hauls in the surf zone at four locations spaced haphazardly along the beach. For each seine haul, two researchers opened the beach seine parallel to shore in approximately 1.5 m of water. Keeping the weighted line flush with the bottom, we dragged the seine perpendicular to the shoreline until reaching the beach. Fish were then immediately removed from the seine, placed in aerated 1 m x 0.5 m x 0.5 m live wells, identified, enumerated, measured (total and standard length) on glazed (smooth) fish boards, and released alive at the site of capture in accordance University of California Santa Barbara's Institutional Animal Care and Use (IACUC) protocol #943. Any fish that appear to be severely injured, moribund, or that did not recover from the stress of trapping were euthanized using Tricaine methanosulfonate (MS-222), a non-inhaled agents approved in the "AVMA Guidelines for the Euthanasia of Animals: 2013 Edition" for finfish [76].

## Baited remote underwater video (BRUV) surveys

We conducted BRUV surveys following methods modified from Vargas-Fonseca et al. [77] and Borland et al. [25]. Each BRUV consisted of a GoPro Hero2 camera (GoPro Inc., San Mateo, California, USA, 2020) mounted on a five kg weight with a line and float attached for ease of retrieval. We then attached a bait bag containing ~152 g of frozen squid (*Loligo* sp.) to the weight with a PVC pipe, positioning it one meter in front of the camera. Snorkelers deployed three haphazardly spaced BRUV units along the outer edge of the surf zone at a depth of greater than two meters within two hours of low tide after conducting sein hauls, except for at sites where sufficient personnel allowed for concurrent sampling. We deployed each BRUV for one hour, producing three hours of video per beach. We reviewed videos to determine fish abundance, species richness, and community composition, using the *MaxN* statistic, the maximum number of individuals of one species in one frame during the hour-long footage [78].

## Environmental DNA (eDNA) surveys

We collected three replicate 0.5 L samples of seawater (herein referred to as sample replicates) using sterile collapsible enteral feeding bags at each site. We then gravity filtered samples through 0.2 μm Sterivex filters following the methods of Gold et al. [79] (See Supplement for detailed methods description), storing filters at -20˚C prior to extraction via a modified Qiagen DNAeasy Blood and Tissue kit (Qiagen Inc., Gernnmantown, MD, USA) [80]. Four field blanks consisting of 1 L of tap water were filtered in the field to serve as field and extraction controls. We amplified eDNA samples in triplicate using both *12S* MiFish Universal teleost (MiFish-U) and elasmobranch (MiFish-E) primer sets [81], and then prepared sequencing libraries preparation followed Gold et al. [79] using Nextera Unique Dual Indices (Illumina, San Diego, CA, USA). Each unique PCR reaction included both positive and negative PCR controls; negative controls substituted molecular grade water in place of the DNA extraction, and we used either

American alligator (*Alligator mississippiensis*) or dromedary (*Camelus dromedarius*), species non-native to California, for positive controls. In total, 2 positive controls, 5 PCR negative controls, and 4 field blanks were sequenced. We pooled all samples in equimolar concentrations by primer set except negative controls (5μL were added given no quantifiable DNA), resulting in a MiFish-U and a MiFish-E library which were separately sequenced on NextSeq PE 2 x 150 bp mid-output at the Technology Center for Genomics & Bioinformatics at the University of California–Los Angeles (UCLA) with 20% PhiX added to both sequencing runs.

## Reference barcode generation

To supplement the California Current Large Marine Ecosystem reference database [79], we generated a MiFish 12S Universal Teleost specific reference sequence for white seabass (*Atractoscion nobilis*). Tissue was acquired from the California Current Cooperative Fisheries Investigation collections and extracted using a Qiagen DNAeasy Blood and Tissue kit (Qiagen Inc., Gernmantown, MD, USA). Tissue was amplified following the same PCR protocol for MiFish 12S Universal Teleost primer set described above. PCR products were purified using ExoSA-P-IT (Affymetrix, Cleveland, OH, USA) and Sanger sequenced in both directions using Big-Dye chemistry (Applied Biosystems Inc, Foster City, CA, USA) at Laragen Inc., (Culver City, CA, USA) following Gold et al. [79].

## eDNA bioinformatics

We processed the resulting eDNA metabarcoding sequences using the *Anacapa Toolkit* (version 1) (Curd et al., 2018) [119], conducting quality control, amplicon sequence variant (ASV) parsing, and taxonomic assignment. Taxonomy was assigned using the Bayesian Lowest Common Ancestor classifier [82] and a curated reference database composed of fishes from the California Current Large Marine Ecosystem supplemented with the generated *Atractoscion nobilis* sequence following [79]; See detailed description in supplement. We processed each sequencing library twice using two different barcoding reference libraries. First, to assign taxonomy to marine mammalian and avian species, we used the *CRUX-generated-12S* database, comprised of reference barcodes for all publicly available *12S* barcodes. Second, we used a curated metabarcoding database specific to California coastal marine fish to generate taxonomic assignments for fishes. We employed a Q score cutoff of 30 and Bayesian taxonomic cutoff score of 60 following the methods of Gold et al. [79]. The resulting taxonomic tables were transferred into *R* for further processing (R Core Team, 2020a). Reads from multiple ASVs with identical assigned taxonomic paths were summed together following the methods of Gold et al. [79]. For example, both ASV 1 and ASV 3 were assigned to *Atherinops affinis* with 1.5 million reads and 1.4 million reads respectively and were thus all reads assigned to both ASVs were summed.

We employed a multifaceted decontamination approach described in Gold et al. [68] developed by Kelly et al. [65] to remove field contamination, lab contamination, and index hopping [65, 83–85]. Through the decontamination process we implemented a hierarchical site occupancy modeling framework to distinguish occupancy rates across multiple sample bottle and technical replicate detections [65]. Only ASVs detected in at least two technical replicates in a site were kept. From these processes, we obtained decontaminated eDNA species-by-sample community tables with counts of total sequence reads. We then summed the total reads of ASVs by assigned taxonomy including multiple ASVs from the two MiFish markers employed (e.g. summed 41 ASVs assigned to Black surfperch *Embiotica jacksoni*). From these processes, we obtained decontaminated eDNA species-by-sample community tables with counts of total sequence reads. We note that this approach does not use sequenced negative controls or field

blanks to correct reads as previous work has demonstrated that this frequently removes the most abundant ASVs which arise as a result of index hopping [65].

We transformed the eDNA read counts into eDNA index scores according to Kelly et al. [65], which normalizes the read count per technical PCR replicate per species. This index was computed by first calculating the proportional relative abundance of each species in each technical PCR replicate. The relative abundance was then divided by the maximum relative abundance for a given species across all samples (e.g. highest observed abundance for Northern Anchovy is 15% of all total reads which serves as the denominator), yielding the eDNA index score, which ranges from 0 to 1 and allows for comparisons of relative abundance for specific taxa across samples. The goal of applying this transformation is to account for the effects of amplification bias across taxa [86–88].

We acknowledge that such an approach loses information about relative abundance between taxa in a sample. In particular, this approach likely over-inflates the abundance of rare taxa [88]. However, recent metabarcoding frameworks have highlighted that the compositional nature of metabarcoding [89] alongside species-specific amplification biases [87, 90], impair our ability to make adequate inferences from raw sequence counts or proportions between taxa without ancillary information on either the underlying target taxa abundances or amplification efficiencies [87, 90–93]. Thus, in the absence of such information, we follow the conservative methods advised by Kelly, Shelton, and Gallego [88] and employ the eDNA index, erring on the side of amplification efficiency bias being the largest contributor to observed differences in read counts across fish species from eDNA samples [72, 87, 94].

## Data analysis

To explore the relative efficacy of seines, BRUV, and eDNA surveys for characterizing surf zone fish communities, we compared the total number of teleost and elasmobranch species identified by each method using the *phyloseq* (version 1.28.0) and *vegan* packages (version 2.5–7) [95, 96] in *R* (version 3.6.1 [97]). Comparisons were made in two ways: 1) all detected fish taxa and 2) only surf zone fish taxa. Surf zone taxa were determined using habitat descriptions from FishBase.org and the literature [4, 98–100] (S2 Table in S2 File). We determined and visualized the overlapping and unique fish species detected by each survey method across all 18 sites using the *VennDiagram* package (version 1.6.20) [101], comparing species richness of each method using Analysis of Variance (ANOVA) and post-hoc Tukey tests using the *vegan* package [96].

To examine survey method performance on a site-by-site basis, we calculated and compared the overlap of presence/absence site-species detections [97, 102]. Here, we define a site-species detection as the detection of a species at a given site (e.g., Top smelt detected at Bechers Bay). Comparisons of site-species detections were conducted for both the surf zone fishes and all fishes, observed by seine, BRUV, and eDNA, respectively. We estimated sample coverage, the fraction of the total incidence probabilities of the discovered species for a set of sampling units, from rarefaction and extrapolation models for species richness (Hill number q = 0) for each method using the *iNext* package (version 2.0.20) [103].

To determine whether the presence or absence of a species is a true reflection of biological reality or due to issues in the sampling process. [104, 105], we also conducted a site-occupancy analysis of species detections at each site following the methods of Chambert et al. [85] as implemented by Kelly et al. [65, 68]. We note this site occupancy analysis is separate from the method implemented in the decontamination process and was conducted on the final quality controlled data set. The binomial model yields the likelihood that a taxon detected is truly present in the sample. The model, implemented in Stan for *R* [104], depends upon three

parameters: 1) the commonness of a taxon in the dataset (denoted $P_{si}$), 2) the probability of a detection given that the taxon is truly present (true positive; denoted $P_{11}$), and 3) the probability of a detection given that the taxon was not truly present (false positive; denoted $P_{10}$). The probability of occurrence function used was the following:

$$Probability\ of\ Occurrence = \frac{p_{si}*p_{11}{}^{N}*(1-p_{11})^{K-N}}{p_{si}*p_{11}{}^{N}*(1-p_{11})^{K-N} + (1-p_{si})*p_{10}{}^{N}*(1-p_{10})^{K-N}}$$

Where K is the number of samples taken within a site and N is the number of species detections within a site (See Supplemental methods for detailed description). For each species we calculated the number of detections out of the number of replicate surveys taken at each site.

We emphasize that the true occupancy of any given species at any site is unknown. Here, we are estimating the true positive ($P_{11}$) and false positive ($P_{10}$) of species being present at a given site using detections across repeated sampling events. Importantly, we make the explicit assumption that any detection of a species by a method is a real detection of that species.

In addition to probability of occurrence we also calculated the mean sensitivity, the proportion of true positive detections correctly identified as positive using the following equation for each species:

$$Sensitivity = \frac{p_{11}}{p_{11} + p_{10}}$$

We also calculated the mean specificity, the proportion of true negative detections correctly identified as negative, using the following equation for each species:

$$Specificity = \frac{1 - p_{10}}{(1 - p_{10}) + (1 - p_{11})}$$

We then compared the probability of occupancy, mean sensitivity, and mean specificity of each method across all species detected [106]. We further compared differences in the eDNA-derived probability of occurrence of surf zone and non-surf zone associated species to test if occupancy rates are a potential function of transport dynamics.

To analyze differences in the composition of surf zone fish detected among methods and across sites, we conducted a PERMANOVA and companion multivariate homogeneity of group dispersions on Jaccard-Binary dissimilarity indices based on presence/absence data using the *adonis* and *betadisper* functions in the *vegan* package [96]. The PERMANOVA was conducted using the following model:

$$Detection \sim Survey\ Method + Site.$$

We excluded the Soledad site on Santa Rosa Island given the lack of a BRUV survey. We further visualized community beta diversity among sampling methods using a constrained canonical analysis of principal components (CAP) through the *vegan* package [95, 97].

Lastly, to assess the ability of eDNA to capture relative abundance, we compared mean eDNA index scores to both the average catch counts per seine as well as average *MaxN* counts per BRUV station using species-specific linear regressions. Similarly, we compared BRUV-derived average *MaxN* counts against average seine counts. We focused our analyses on species detected jointly by each method at three or more sites. We note that comparing uncorrected compositional results (eDNA metabarcoding data) to estimates of absolute abundance (BRUV *MaxN* and seine counts) is inherently flawed [31, 87]. We present such results here to highlight the caveats of such approaches and discuss their merits (See Discussion).

## Results

Our beach seine surveys captured a total of 1,359 individuals in 72 hauls across all sites (4 hauls per site). Seven of the 72 hauls produced 0 individuals. In total, seining detected 24 species of fish from 24 genera, 13 families, and two classes (S3 Table in S2 File). On average, we captured 4.0 species (± 2.5 standard deviation, range 0–9), and 75.5 ± 82.8 individuals per site (range 0–325 individual fishes).

Our BRUV surveys detected a total of 1,114 individual fishes in 51 BRUV deployments (3 replicate deployments per site). In total, BRUV surveys detected 30 species, 30 genera, 21 families, and two classes (S4 Table in S2 File). An average of 6.3 ± 3.2 species (range 2–16 species) and 65.5 ± 65.5 total individuals (range 13–236 individuals) were recorded per site.

We successfully generated a reference barcode sequence for *Atractoscion nobilis* (S5 Table in S2 File).

Sequencing of the 54 eDNA samples yielded a total of 4,839,3365 MiFish elasmobranch reads and 16,835,329 MiFish teleost reads that passed the initial Illumina NextSeq quality controls across all samples. After decontamination and site occupancy modeling, we retained 3,652,862 reads and 915 ASVs from MiFish Elasmobranch primer set and 14,159,828 reads and 1,967 ASVs from MiFish Teleost primer set, representing 89 species of fish from 79 genera, 46 families, and two classes across sites. On average we observed 34.7 ± 11.1 SD species per site (range 17–59 species) (S1 Fig in S1 File). We note that in addition to fish we detected 8 species of birds and 5 species of marine mammals not discussed here.

Furthermore, six species of fish were detected in our field and negative controls when sequenced, despite the fact that no bands were visualized in PCR products during gel electrophoresis. Five of these six species were in the top 7 most abundant ASVs and in total represented 0.05% of all reads across all samples (n = 89,521). Given the prevalence of these species across all samples and technical replicates we did not use blanks for decontamination purposes [65].

### Species assemblages characterized by each method

We found variable agreement in the assemblages of species captured by each survey method across all 18 sites (Fig 2 & S2 Fig in S1 File). Seine and BRUV captured distinct, but overlapping surf zone fish assemblages, sharing only 50% (18/36) of fishes species. Seine surveys detected 6 species of fishes not observed in BRUV surveys, including 2 species of croakers (Family Sciaenidae), 2 species of surfperches (Family Embiotocidae), and two planktotrophic coastal-pelagic species (families Clupeidae and Atherinidae). In contrast, BRUV surveys detected 12 fish species not observed in seines, including 3 species of elasmobranchs, 6 species of rocky reef associated species, and 2 coastal-pelagic predator species.

In contrast, eDNA detected the majority (88.9%, 32 out of 36) of species found in seine and BRUV surveys (Fig 2). Similarly, when only focusing on surf zone fish (S2 Table in S2 File), eDNA detected 93.1% (27 out of 29) of species detected in seine and BRUV surveys (S2 Fig in S1 File). eDNA methods failed to detect four species found in the seine and BRUV surveys including three species of surfperch, the most abundant and widespread family (*Embiotocidae*) detected in the seine surveys. Undetected species include the walleye surfperch *(Hyperprosopon argenteum)*, silver surfperch *(Hyperprosopon ellipticum)*, barred surfperch *(Amphistichus argenteus)* and kelp pipefish *(Syngnathus californiensis)*. However, eDNA surveys detected 57 fish species not detected in seine or BRUV surveys (S6 & S7 Tables in S2 File), including 15 surf-zone associated species and 42 species more typically associated with reef and pelagic habitats (S2 Table in S2 File). Thus, eDNA had high overlap with both BRUV and seine surveys in addition to capturing additional surf zone and nearshore marine fishes.

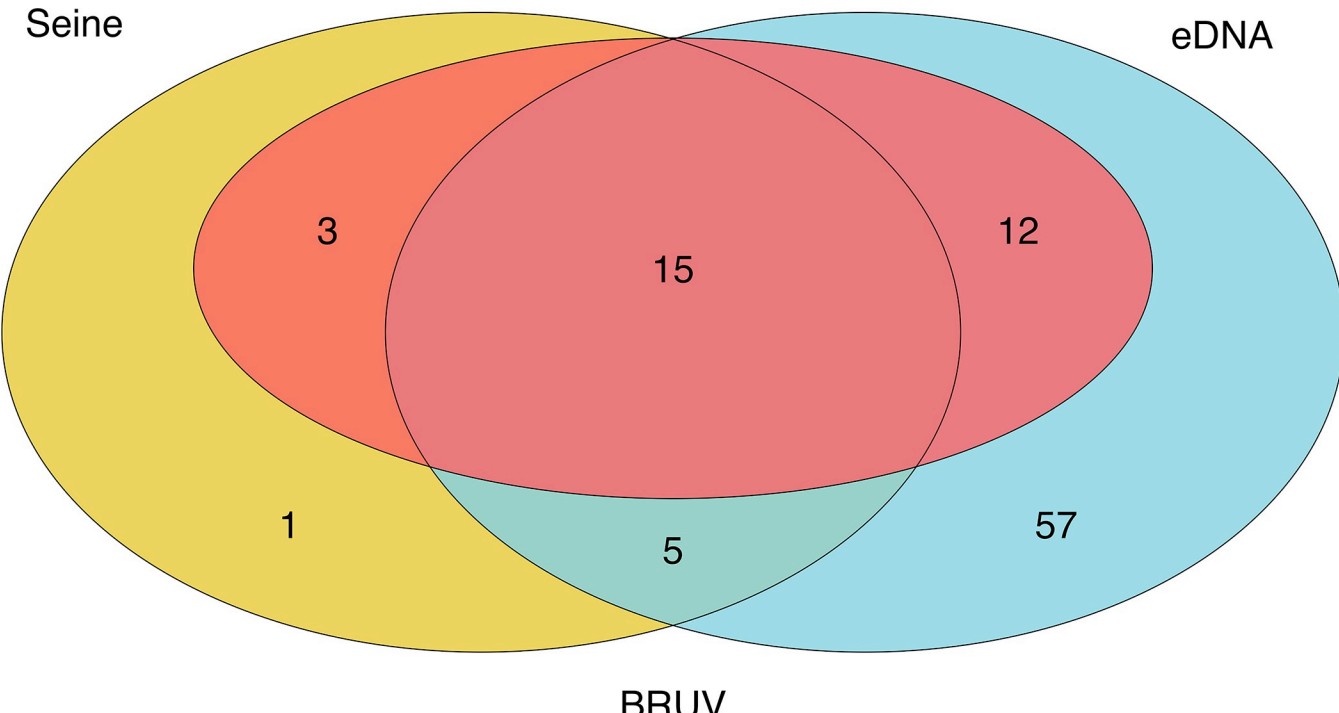

**Fig 2. Venn diagram of eDNA, seine, and BRUV species detections.** Environmental DNA methods captured the majority (30/36) of fish species detected by both BRUV and seine surveys, only failing to identify six fish species found in the other two survey methods. In addition eDNA identified 58 additional fish species missed by seine and BRUV methods. In contrast, BRUV and seine surveys only captured 50% of species detected by both methods, showing strong difference in the species detected by each method. This was largely driven by the unique detection of elasmobranchs as well as nearshore pelagic and rocky reef carnivorous fishes in BRUV surveys compared to the unique detection of surfperches (Family Embiotocidae), grunts (Family Sciaenidae), and planktivorous nearshore pelagic species in seine surveys.

Composition of detected taxa varied significantly among survey methods (Fig 3 and S3 Fig in S1 File; CAP ANOVA p < 0.001) driven by biases in detection of specific taxa. Seines and BRUVs commonly detected barred surfperch (*Amphistichus argenteus*), whereas eDNA only could not resolve surfperches below family level. Similarly, eDNA and BRUV surveys more frequently detected leopard shark (*T. semifasciata*) and California bat ray (*M. californica*) compared to seine surveys. In contrast, eDNA detected many more species than BRUVs or seines, including opaleye (*Girella nigricans*), northern anchovy (*Engraulis mordax*), giant kelpfish (*Heterostichus rostratus*), and dwarf perch (*Micrometrus minimus*) (Fig 3).

In total, survey method explained 42.6% of the total variation observed in the composition of detected taxa, while site explained an additional 28.3% (PERMANOVA p <0.0001). We found no significant difference in homogeneity of dispersions across methods or sites (betadisper > 0.05) (S8 Table in S2 File). We also found similar differences in fish communities between survey methods when we limited our comparisons to only taxa observed by both visual and eDNA methods. Survey method explained 33.2% of the total variation observed in the composition of detected taxa, while site explained an additional 36.8% (PERMANOVA, p< 0.001). However, eDNA had significantly lower dispersion than seines across all sites (homogeneity of dispersions p< 0.001) (S9 Table in S2 File).

### Detection rates of species across methods

Detection rates of species also differed significantly among survey methods (Fig 4 & S4-S8 Figs in S1 File) with eDNA having a significantly higher sensitivity (98.5%) than both seine (96.7%)

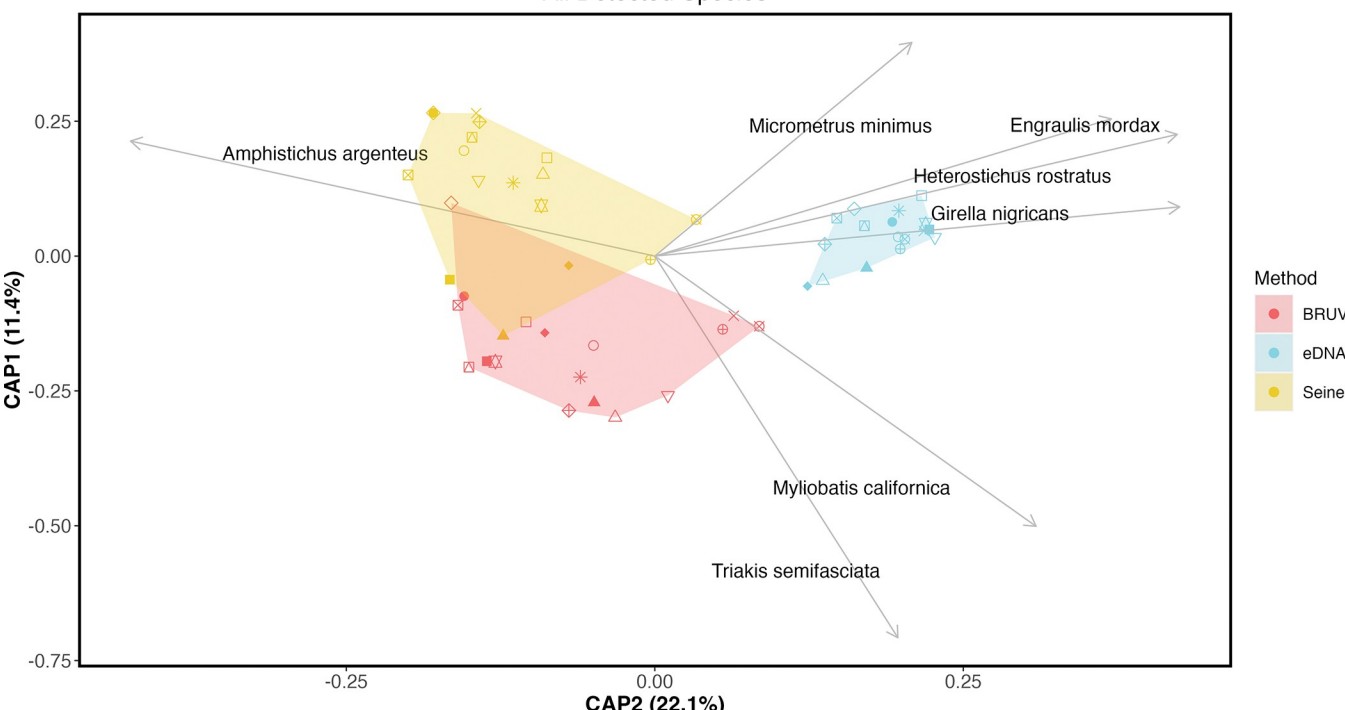

**Fig 3. Constrained analysis of principal components.** Constrained Analysis of Principal Components (CAP) analysis was conducted on Jaccard binary dissimilarities of fish assemblages of all species detected across surveys. Survey method explained 42.6% of the total variation observed in the composition of detected taxa while site explained an additional 28.3% (PERMANOVA p <0.0001). We found no significant difference in homogeneity of dispersions across sites (betadisper > 0.05). BRUV and eDNA approaches more frequently detected leopard sharks (*Triakis semifasciata*) and California bat ray (*Myliobatis californica*) compared to seine surveys. Both seine and BRUV surveys detected Barred surfperch (*Amphistichs argenteus*) while eDNA methods could only achieve family level resolution for this taxon. eDNA approaches more consistently detected opaleye (*Girella nigricans*), northern anchovy (*Engraulis mordax*), giant kelpfish (*Heterostichus rostratus*), and dwarf perch (*Micrometrus minimus*).

and BRUV (96.2%) surveys across all taxa (ANOVA, p< 0.0001). Likewise, eDNA had significantly higher probability of occupancy (47.2%) at the site level than both seine (24.9%) and BRUV (28.6%) surveys (ANOVA, p< 0.0001) as well as having significantly higher specificity (71.4%) than seines (66.4%) (ANOVA, p = 0.01). However, we observed no difference in specificity between BRUV (69.1%) and eDNA or seine surveys at the site level (ANOVA, p >0.5) (Fig 5). Furthermore, we found no significant difference in probability of occupancy for species known to inhabit surf zone habitats (52.4%) than non-surf zone associated species (42.6%) detected with eDNA methods (ANOVA, p = 0.06) (S8 Fig in S1 File).

The three methods yielded different levels of detection both overall and of individual species of surf zone fish. Our eDNA samples more consistently detected 96.9% (31/32) of all species jointly observed by either BRUV or seines. Leopard shark (*Triakis semifasciata*) was the only species more frequently detected with BRUV (15/18) than eDNA methods (14/18). When grouping surfperch species with identical 12S reference barcodes at the family level, eDNA detected surfperches as frequently as seine surveys (17/18) (Fig 4). When grouping all *Syngathus* sp. pipefish species with identical 12S reference barcodes at the genus level, eDNA detected *Syngathus* sp. More frequently than seine and BRUV surveys.

Comparing only BRUV and seine surveys, our BRUV surveys detected elasmobranchs and flatfishes (Families Pleuronectidae and Paralichthyidae) more frequently than seine surveys. In contrast, seine surveys more frequently detected barred surfperch (*Amphistichus argenteus*),

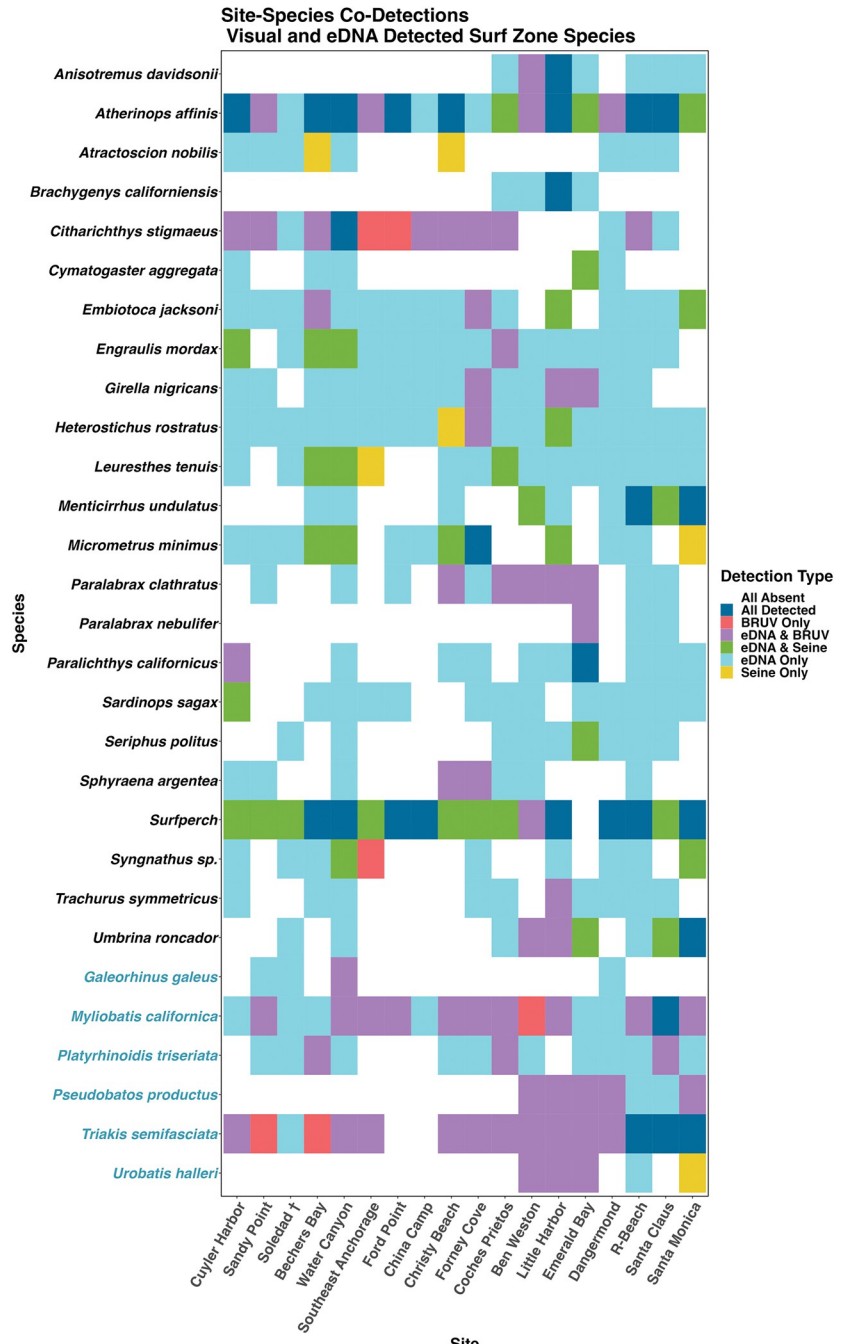

**Fig 4. Heatmap of surf zone fishes jointly detected between surveys.** Teleost species in black font and elasmobranch species in blue font. Environmental DNA approaches more frequently detected 26 of 27 known surf zone species detected by either BRUV or seine surveys. Only Leopard shark *Triakis semifasciata* was more frequently detected by BRUV surveys. Aggregating surfperches with identical 12S barcodes to family level assignment (surfperch), eDNA detected surfperches at the same rate at seines. Aggregating to genus level assignment, eDNA detected *Syngathus* sp. More frequently than either BRUV of seine surveys.

walleye surfperch (*Hyperprosopon* argenteum), California corbina (*Menticirrhus undulatus*), northern anchovy (*Engraulis mordax*), giant kelpfish (*Heterostichus rostratus*), and kelp pipe-fish (*Syngnathus californiensis*) than BRUV surveys.

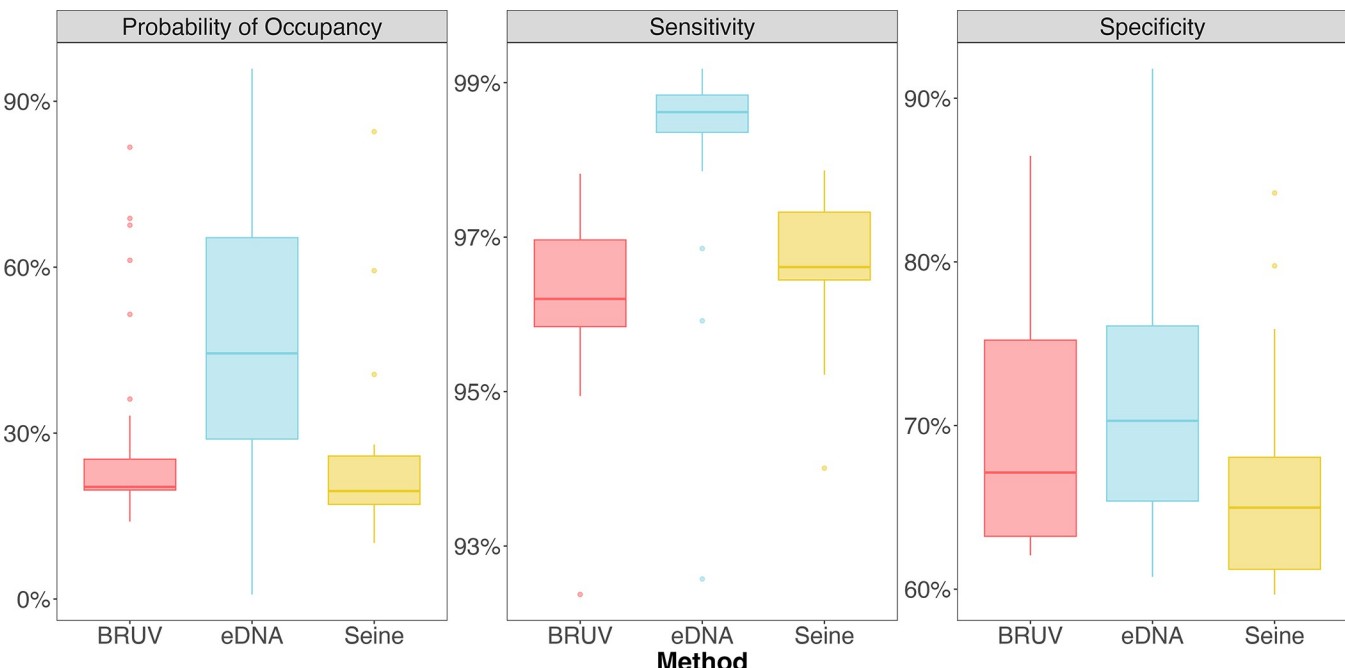

**Fig 5. Probability of occupancy, specificity, and sensitivity of eDNA, seine, and BRUV surveys.** Environmental DNA surveys had higher probability of occupancy and sensitivity than BRUV and seine surveys. eDNA had significantly higher specificity than seine surveys. We found no difference in specificities between BRUV and eDNA and seine surveys. Probability of occurrence is a measure of how likely a species is present at a site as a function of the commonness of the species as well as the true positive and false positive detection rates of the method surveyed. Sensitivity is the proportion of true positive species detections correctly identified as true positive detections. Specificity is the proportion of true negative species detections identified as negative detections.

Across all sites, eDNA had higher sample coverage estimates (98.6%) than both BRUV (89.6%) and seine (85.2%) surveys (Fig 6). From species rarefaction curves of all species surveyed at the site level, we estimate that both BRUV and seine surveys would have to be conducted at more than 100 sites to achieve similar sample coverage estimates as eDNA at the 18 sites surveyed here.

## Comparisons of relative abundance among survey methods

Estimates of relative abundance varied significantly among the three survey methods and were generally not correlated. We found a significant positive relationship between BRUV *MaxN* values and seine counts ($R^2$ = 0.31, p = 0.031, S11 Table in S2 File, S9 Fig in S1 File) for only one species, topsmelt (*Atherniops affinis*). Likewise, there was a significant positive relationship between seine counts and eDNA index scores for only two species, topsmelt, ($R^2$ = 0.27, p = 0.03, S12 Table in S2 File, S10 Fig in S1 File), and California corbina, *Menticirrhus undulatus* ($R^2$ = 0.91, p < 0.001, S12 Table in S2 File). Similarly, there was a significant positive relationship between BRUV *MaxN* and eDNA index for three species (kelp bass, *Paralbrax clathratus*, shovelnose guitarfish, *Psuedobatos productus*, and round stingray *Urobatis halleri*) (respective $R^2$: 0.33, 0.41, and 0.94, p < 0.01, S13 Table in S2 File, S11 Fig in S1 File).

## Discussion

Despite extreme methodological differences, seine, BRUV, and eDNA surveys captured largely overlapping, but distinct fish assemblages in surf zone habitats with notable taxonomic biases. Seines more consistently detected surfperches, including the most abundant fished species, barred surfperch (*Amphistichus argenteus*) while BRUV surveys efficiently revealed larger

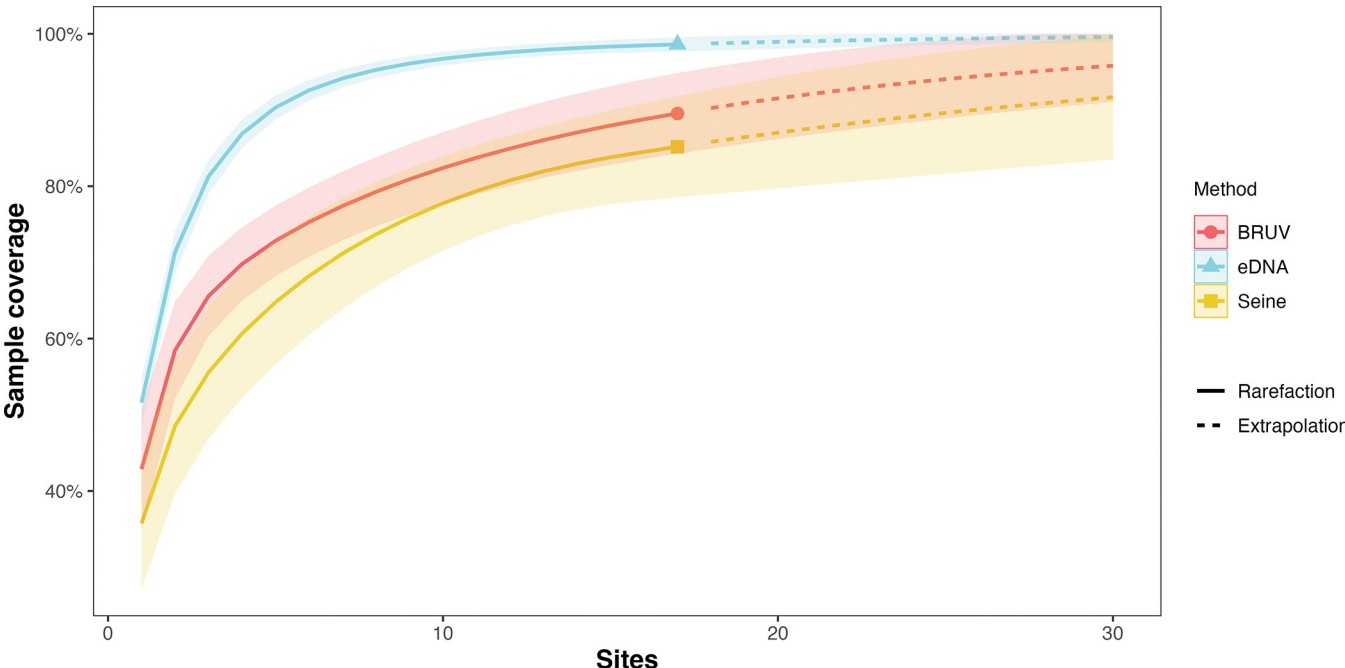

**Fig 6. Sample coverage estimates of eDNA, seine, and BRUV surveys.** Across all sites, environmental DNA surveys had an estimate sample coverage of 98.9%, higher than the sample coverage of BRUV (89.6%) and seine (85.2%) surveys. Shaded area represents 95% confidence intervals. Sample rarefaction curves across sites suggest BRUV and seine surveys would have to be conducted at more than 100 sites to achieve similar sample coverage estimates to eDNA surveys conducted at the 18 sites surveyed here.

predatory species, particularly elasmobranchs as previously documented [107]. eDNA captured the highest species richness of all three methods, including the majority of species detected by seine and BRUV surveys (32/36). The mismatch in fish assemblages sampled by each method made comparisons of relative abundance difficult, highlighting a key challenge of comparing survey methods [43].

Importantly, similar to other studies (see [32, 33, 35, 41, 43, 68, 108] among others), we found that eDNA analysis had higher sensitivity than the two traditional methods, and more frequently detected nearly all jointly observed species at a given site. Our results suggest that seine, BRUV, and eDNA approaches are complementary and their use in tandem provides the most accurate characterization of surf zone fish communities. Recent studies using two of these three methods reached similar conclusions [107–109].

## Species assemblages characterized by each method

Only half of fish species detected by seine and/or BRUV surveys overlapped (18/36) indicating that these methods target different species assemblages. Compared to BRUV surveys, seine surveys captured additional surfperches and croakers associated with surf zone habitats as well as planktivorous coastal pelagic species. In contrast, BRUV surveys detected a greater number of elasmobranch and rocky reef species, particularly carnivores and scavengers, suggesting that fish are attracted from adjacent habitats to the bait, or our current understanding of species' surf zone habitat utilization is limited. Combined, our results align well with previous findings from tropical shorelines indicating that BRUV and seines capture distinct, but overlapping fish assemblages in surf zone habitats [107].

Our finding that eDNA approaches detected nearly 90% (32 out of 36) of fish species observed using seine and BRUV methods, with higher overlap in detected fish assemblages.

Importantly, eDNA approaches also captured an additional 15 surf zone species not observed by our seine or BRUV methods, including the federally listed northern tidewater goby (*Eucyclogobius newberryi*) and commercially-fished species of management concern, such as the flathead grey mullet (*Mugil cephalus*), black croaker (*Cheilotrema saturnum*), white croaker (*Genyonemus lineatus*), and Pacific sanddab (*Citharichthys sordidus*) [4]. Furthermore, eDNA detected a wide array of elasmobranchs that are typically underrepresented in most traditional sampling approaches [108, 110–112] including angel shark (*Squatina californica*), horn shark (*Heterodontus francisci*), California butterfly ray (*Gymnura marmorata*), and broadnose sevengill shark (*Notorynchus cepedianus*). As such, eDNA should be viewed as a valuable complement both seine and BRUV surveys.

The failure of eDNA to detect four common species captured by seine and BRUV surveys was due to the limitations of the 12S MiFish-U primers, particularly for the surfperches and pipefishes [79]. Both Embiotocidae and Syngnathidae are diverse, recent radiations [113, 114] and the MiFish-U primers perform poorly in such cases, such as rockfish in the genus *Sebastes* [115]. Failure to detect three of six surfperch species and is likely a result of insufficient genetic variation within the 12S gene region bounded by the MiFish 12S primer set, leading to many surfperches only being resolved at higher taxonomic ranks (e.g. Embiotocidae) [79]. Importantly, we note all three species had shared corresponding 12S reference barcodes [79]. Likewise, kelp pipefish (*Syngnathus californiensis*) and bay pipefish (*Syngnathus leptorhynchus*) have identical 12S reference barcodes and thus could not be resolved to species level [114]. However, given that we were able to resolve surfperches to family level and pipefishes to genus level resolution with eDNA suggests we were able to capture shed DNA from these taxa and that the application of alternative primer sets could have led to improved results [79, 114, 115].

Although the MiFish-U primer set is imperfect, our results demonstrate the value of generating reference barcode sequences for all taxa in the study system. At the request of an anonymous reviewer, we generated a MiFish-U reference sequence for white seabass (*Atractoscion nobilis*) which allowed us to successfully detect the species at multiple sites unlike in previous applications of eDNA in coastal Southern California waters [52, 67, 79, 116, 117]. Without the barcode, the species could only be resolved to Family level and was not included in analyses. These results add further support to the importance of reference barcode generation for successful eDNA efforts [68, 118, 119].

eDNA captured a strong signature of surf zone fish assemblages including an additional 15 species of surf zone fishes not observed by seine and BRUV approaches, highlighting the utility of eDNA biomonitoring to improve estimates of total fish diversity in coastal monitoring surveys. eDNA also detected an additional 42 native coastal marine fishes not detected by our seine and BRUV surveys (S6, S7 Tables in S2 File). Although many of these species are unlikely to inhabit surf zone habitats directly [4, 100, 120], our study beaches were adjacent to rocky reef kelp forests, rocky intertidal habitats, and estuaries. Our detections of additional native fish species highlight the capacity for movement of both fish and eDNA across pelagic and inshore habitats [63]. Given the potential for transport on the scale of tens to thousands of meters, the detection of fishes from adjacent habitats in eDNA samples is to be expected [66], thus highlighting a potential shortcoming of eDNA approaches, and the need for better understanding of spatial and temporal variability in the dispersal of eDNA within and across ecosystems. Despite the need to better characterize the fate and transport of eDNA, our results still demonstrate that such eDNA approaches can be highly informative of surf zone communities [52], particularly on longer open coast beaches that are not located adjacent to rocky subtidal or intertidal habitats.

## Detection rates of species across methods

In addition to the differences in fish assemblages captured by each method, we found substantial differences in the detection frequency of jointly observed species across sites between these methods. Overall, we found that eDNA had higher frequency of detection of nearly all species (31/32) jointly detected by either of the seine and BRUV methods (S6, S7 Tables in S2 File). This higher rate of detection also resulted in eDNA having significantly higher sensitivity than both seines and BRUV surveys. Furthermore, results from species rarefaction curves suggest that eDNA surveys capture a larger proportion of the total fish diversity across sites than seine and BRUV surveys. Importantly, our results suggest that additional BRUV and seine surveys should be deployed across more sandy beach sites rather than additional deployments at the same site to maximize fish diversity across the region. In contrast, our results suggest that the current eDNA deployment of three sample replicates with three technical replicates was sufficient to adequately capture diversity across the region, providing a baseline sampling regime for future eDNA deployments for monitoring fish diversity in surf zone ecosystems.

One possible explanation for the differences in site-species detection frequency across methods is poor taxonomic resolution or erroneous assignment across methods. The *Anacapa Toolkit* provides confidence scores around each taxonomic rank of assignment, providing information on the accuracy of eDNA identifications [119]. However, such confidence scores are not readily available for data from seine and BRUV surveys, where taxonomic identification depends on the presence of easily observed morphological characteristics and the resolution of video still captures. For example, topsmelt (*Atherinops affinis*) and California grunion (*Leurethes tenuis*) are morphologically very similar, with the potential for misidentification, particularly under low visibility conditions for BRUV surveys.

The variation in temporal and spatial scales sampled by each of the three survey methods may also drive differences in site-species detections [26, 63, 67, 70, 71, 107, 108]. Beach seines survey a small spatial area (here 15.3 m x 1.8 m x 2m) at 0 to 1.5 m depth at a single instantaneous snapshot of sampling [31, 120]. In contrast, BRUV units were deployed for an hour at 2–3 m depth and likely attracted species across tens to hundreds of square meters [26, 73, 107, 108, 121]. Although the spatial and temporal scales of eDNA methods in marine systems are still an active area of research, previous studies have found that eDNA integrates across spatial scales from 50–1,000 meters and degrades *in situ* between 2 and 12 hours, although laboratory experiments suggest degradation rates on the order of days [53, 63, 67, 71, 122]. Thus, the ecological integration time of each of these surveys is substantially different and likely contributes to the differences we observed in species detections [43].

Differences in species detection among methods are also likely driven by the dynamics of eDNA. eDNA shedding rates can vary among [123] and within species [124], driven by differences in physiology and behavior. Increased shedding rates result in higher eDNA detection probabilities, thus biasing which species are successfully detected within surf zone ecosystems. For example, eDNA methods have the potential to be biased during spawning events when high DNA concentrations are released [125]. Likewise, the interaction between high water transport within and potentially variable degradation rates across species or environmental conditions (temperature, UV, etc.) could influence detection probabilities [64, 70, 126].

We found that eDNA captured a wide variety of species not typically associated with surf zone habitats, suggesting transport of eDNA from offshore and other intertidal habitats and some level of spatial integration of eDNA measurements. Interestingly, we found that species known to inhabit surf zone habitats had similar probability of occupancy than species known to associate with further offshore habitats. Previous work has found that eDNA signatures were able to distinguish surf zone and adjacent subtidal kelp forest ecosystems from

differences in fish assemblage composition as well as relative abundance estimates [52]. However, our results suggests that eDNA detection is not biased towards species known to generally inhabit surf zone habitats. Our observational study design precluded directly testing whether species recently inhabiting the surf zone upon sample collection had higher rates of detection as many of the species known to inhabit surf zones use such habitats at different frequencies and durations. Future additional research using field based exclusions of specific taxa alongside on eDNA dispersal tracking could allow for modeled adjustment of eDNA data to account for transportation dynamics.

## Relative abundance

Given the observed low overlap in species detections across survey methods, assessing the capability of eDNA approaches to estimate relative abundance was challenging, particularly since eDNA surveys frequently detected a species at multiple sites where seine and BRUV surveys did not detect that species at all. This presents a core challenge of comparing eDNA to capture and visual surveys when the true abundance of species is unknown (Table 1) [43]. However, given that the ability to estimate relative abundance is a function of the ability to detect a given species, our results suggest that eDNA approaches are more sensitive and better suited than capture and visual survey methods to estimate abundance [85, 104, 105]. This result, however, is highly dependent on the ability of eDNA approaches to resolve a given taxa. Here eDNA approaches using the MiFish-U primer set failed to resolve the most abundant surf zone species from both seine and BRUV surveys, the surfperches (Family Embiotocidae).

Recent work from studies with greater survey overlap show promise for estimating relative abundance using eDNA approaches [31, 40, 62, 87, 94, 127], particularly when accounting for the underlying mechanisms that relate observed sequence read counts to the underlying biology and biomass of detected species [72, 90]. Here we note that we did not account for such underlying mechanisms in our comparisons of relative abundance which may explain why we observed such poor correlations. Such poor correlations between our simple eDNA index transformation and seine and BRUV counts are expected given the compositional nature of metabarcoding [89], as such a transformation does not address the many underlying non-linear factors that affect the relationship between true environmental abundance and observed reads obtained from eDNA metabarcoding sequencing [87, 90, 92]. Future work should account for the suite of mechanisms that affect observed sequence reads including transport, residence time, and variation in species specific shedding and degradation rates of eDNA [72, 94, 126, 128]. Importantly such efforts must account for the role of amplification efficiency for biasing metabarcoding results using joint models of metabarcoding data, amplification efficiency estimates, and species specific absolute abundance estimates [88, 90, 92, 127].

## Choosing a survey method

All survey methods have biases, and the more a particular survey method is used allows the determination of such biases. For example, diver avoidance behavior is a well-established bias of visual SCUBA surveys [19, 27–29]. Likewise, results of this study showed that each method had distinct advantages and disadvantages. BRUVs are more likely to capture large mobile species than seines, and eDNA captured more total diversity than BRUVs or seines. As such, method selection will largely be a function of the goals of a study, and whether detection of specific taxa or total diversity is a priority.

However, an important consideration when employing eDNA or BRUV data compared to seine surveys (without photographic documentation of hauls) is that the DNA sequences and ASV tables generated by eDNA and the video footage produced by BRUVs are permanent

records of what was present at a particular time [32] (Table 1). For eDNA in particular, as reference databases are improved, eDNA sequence data can be reanalyzed to test for the presence of previously missed or poorly resolved taxa, e.g. white seabass. In addition, bio-archived eDNA samples or extractions can be revisited for future resequencing and management and biomonitoring applications (e.g., species invasions) [129]. The ability for future analyses of a given ecosystem at a specific time highlights the advantages of applying multiple approaches, where eDNA can provide robust and accurate taxonomic information that can be updated over time while carefully deployed stereo-video approaches (not deployed here given challenging surf conditions) and seine hauls can provide size structure and biomass estimates with demonstrated utility [26, 107].

## Conclusion

There is a growing need to survey threatened surf zone and beach ecosystems in the face of global change [11]. Our results suggest that seine, BRUV, and eDNA approaches are complementary techniques for surveying fish diversity in open coast surf zone habitats. eDNA is a relatively quick, effective, and nondestructive approach to surveying marine wildlife, compared to capture and visual surveys of dynamic surf zone habitat (Table 1). Given the cost effectiveness and ability to automate collection and processing, eDNA methods could provide an approach to increase the scope and scale of surf zone ecosystem monitoring across time and space [33, 36]. The ease of sample collection in this challenging habitat could allow researchers, marine resource managers, and community scientists to conduct surveys more frequently and in more places, better characterizing surf zone biodiversity and dynamics [24, 59, 60, 130]. Furthermore, the ability to archive eDNA samples for future use provides an important resource for comparative analyses of ecosystem change [32, 129] and for making use of advances in reference libraries [68, 118].

Although we demonstrated that eDNA provides more robust species detections in surf zone habitats, eDNA cannot provide information on sex ratios or population size structure that can be obtained from seine and BRUV surveys, information critical to resource management [2, 120]. Thus, eDNA cannot be viewed as a wholesale replacement for other survey methods, but instead as a complementary tool for biomonitoring surf zone ecosystems [109]. Nevertheless, adding eDNA surveys to traditional monitoring programs or conducting them on their own when and where other methods are untenable has the potential to greatly enhance our knowledge of surf zone fish communities, providing a new source of comprehensive and detailed information needed for management and preservation of these vital coastal ecosystems in the face of global change.

## Supporting information

**S1 File.**
(DOCX)

**S2 File.**
(XLSX)

## Acknowledgments

We thank the Channel Islands National Park, The Nature Conservancy, UC Natural Reserve System, CSUCI Santa Rosa Island Field Station, USC Wrigley Institute and NOAA Channel Islands National Marine Sanctuary for access and use of facilities. We thank Laura Beresford, Francesca Puerzer, Justin Hoesterey, and Russel Johnson for their assistance in the field. We

thank Beverly Shih, Nikita Sridhar and Lauren Man for their help in library preparation of eDNA samples. We thank Garfield Kwan for generously providing white seabass tissue.

## Author Contributions

**Conceptualization:** Zachary Gold, McKenzie Q. Koch, Nicholas K. Schooler, Jenifer E. Dugan, Robert J. Miller, Henry M. Page, Donna M. Schroeder, David M. Hubbard, Jessica R. Madden, Stephen G. Whitaker, Paul H. Barber.

**Data curation:** Zachary Gold, Stephen G. Whitaker, Paul H. Barber.

**Formal analysis:** Zachary Gold, McKenzie Q. Koch, Nicholas K. Schooler, Kyle A. Emery, Jenifer E. Dugan, Robert J. Miller, Henry M. Page, Paul H. Barber.

**Funding acquisition:** Zachary Gold, McKenzie Q. Koch, Nicholas K. Schooler, Jenifer E. Dugan, Robert J. Miller, David M. Hubbard, Stephen G. Whitaker, Paul H. Barber.

**Investigation:** Zachary Gold, McKenzie Q. Koch, Nicholas K. Schooler, Kyle A. Emery, Jenifer E. Dugan, Robert J. Miller, Henry M. Page, Donna M. Schroeder, David M. Hubbard, Jessica R. Madden, Stephen G. Whitaker, Paul H. Barber.

**Methodology:** Zachary Gold, McKenzie Q. Koch, Nicholas K. Schooler, Kyle A. Emery, Jenifer E. Dugan, Robert J. Miller, Henry M. Page, Donna M. Schroeder, David M. Hubbard, Jessica R. Madden, Stephen G. Whitaker, Paul H. Barber.

**Project administration:** Zachary Gold, McKenzie Q. Koch, Nicholas K. Schooler, Jenifer E. Dugan, Robert J. Miller, Henry M. Page, Donna M. Schroeder, David M. Hubbard, Jessica R. Madden, Stephen G. Whitaker, Paul H. Barber.

**Resources:** Zachary Gold, McKenzie Q. Koch, Nicholas K. Schooler, Jenifer E. Dugan, Robert J. Miller, Donna M. Schroeder, Stephen G. Whitaker, Paul H. Barber.

**Software:** Zachary Gold, McKenzie Q. Koch.

**Supervision:** Zachary Gold, Nicholas K. Schooler, Jenifer E. Dugan, Robert J. Miller, Donna M. Schroeder, Stephen G. Whitaker, Paul H. Barber.

**Validation:** Zachary Gold, McKenzie Q. Koch, Jenifer E. Dugan, Robert J. Miller, Henry M. Page, David M. Hubbard, Paul H. Barber.

**Visualization:** Zachary Gold, McKenzie Q. Koch, Nicholas K. Schooler, Kyle A. Emery, Jenifer E. Dugan, Robert J. Miller, Henry M. Page, Paul H. Barber.

**Writing – original draft:** Zachary Gold, McKenzie Q. Koch, Nicholas K. Schooler, Kyle A. Emery, Jenifer E. Dugan, Robert J. Miller, Henry M. Page, Donna M. Schroeder, David M. Hubbard, Jessica R. Madden, Stephen G. Whitaker, Paul H. Barber.

**Writing – review & editing:** Zachary Gold, McKenzie Q. Koch, Nicholas K. Schooler, Kyle A. Emery, Jenifer E. Dugan, Robert J. Miller, Henry M. Page, Donna M. Schroeder, David M. Hubbard, Jessica R. Madden, Stephen G. Whitaker, Paul H. Barber.

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
