## [Decision Letter · Decision Letter 0]

7 Jul 2022

PONE-D-21-35964Scoping the Line Up: A Comparison of Biomonitoring Methodologies for Surf Zone Fish CommunitiesPLOS ONE

Dear Dr. Gold,

Thank you for submitting your manuscript to PLOS ONE. After careful consideration, we feel that it has merit but does not fully meet PLOS ONE’s publication criteria as it currently stands. Therefore, we invite you to submit a revised version of the manuscript that addresses the points raised during the review process.  Please submit your revised manuscript by August 21, 2022. If you will need more time than this to complete your revisions, please reply to this message or contact the journal office at plosone@plos.org. Please include the following items when submitting your revised manuscript:A rebuttal letter that responds to each point raised by the academic editor and reviewer(s). You should upload this letter as a separate file labeled 'Response to Reviewers'.A marked-up copy of your manuscript that highlights changes made to the original version. You should upload this as a separate file labeled 'Revised Manuscript with Track Changes'.An unmarked version of your revised paper without tracked changes. You should upload this as a separate file labeled 'Manuscript'.

We look forward to receiving your revised manuscript.

Kind regards,

Alejandro Pérez-Matus

Academic Editor

PLOS ONE

Journal Requirements:

5. Please note that in order to use the direct billing option the corresponding author must be affiliated with the chosen institute. Please either amend your manuscript to change the affiliation or corresponding author, or email us at plosone@plos.org with a request to remove this option.

7. Please upload a new copy of Figure 1 as the detail is not clear. Please follow the link for more information: https://blogs.plos.org/plos/2019/06/looking-good-tips-for-creating-your-plos-figures-graphics/" https://blogs.plos.org/plos/2019/06/looking-good-tips-for-creating-your-plos-figures-graphics/

8. We note that Figure 1 in your submission contain map images which may be copyrighted. All PLOS content is published under the Creative Commons Attribution License (CC BY 4.0), which means that the manuscript, images, and Supporting Information files will be freely available online, and any third party is permitted to access, download, copy, distribute, and use these materials in any way, even commercially, with proper attribution. For these reasons, we cannot publish previously copyrighted maps or satellite images created using proprietary data, such as Google software (Google Maps, Street View, and Earth). For more information, see our copyright guidelines: http://journals.plos.org/plosone/s/licenses-and-copyright.

Reviewers' comments:

Reviewer's Responses to Questions

**Comments to the Author**

1. Is the manuscript technically sound, and do the data support the conclusions?

Reviewer #1: Partly

Reviewer #2: Yes

2. Has the statistical analysis been performed appropriately and rigorously? 

Reviewer #1: Yes

Reviewer #2: Yes

3. Have the authors made all data underlying the findings in their manuscript fully available?

Reviewer #1: No

Reviewer #2: Yes

4. Is the manuscript presented in an intelligible fashion and written in standard English?

Reviewer #1: Yes

Reviewer #2: Yes

5. Review Comments to the Author

Reviewer #1: This interesting report compares eDNA to two alternative technologies, hand-operated seining and baited remote underwater video (BRUV), for surveying fish presence and abundance in surf zone of sandy beaches. This is of considerable importance as the surf zone is a difficult habitat to survey by capture and visual methods. Furthermore this is one of a small number of studies that directly compare eDNA to concurrent surveys with other methods, an approach that is essential to advancing adoption of eDNA technology.

I have few major comments/suggestions, plus a number of other points that I think would improve clarity of manuscript.

Major comments

1. I suggest lumping together the seine/BRUV species that cannot be distinguished by eDNA, and then do the analysis. I think this would give a clearer picture of the relative performance of the two methods. By my analysis surfperches (Embocitidae) were 2nd most abundant by reads and 3rd most abundant by detections--it is strange to leave these out of direct comparisons with seine/BRUV, e.g., Fig. 4.

2. Can you find a specimen of white seabass (Atractoscion nobilis) to generate a 12S reference sequence? This is less important given that only 2 specimens were caught, but still would add depth to results.

3. Were there any negative field controls? If not please address. Also I didn't see a statement about results with negative laboratory controls (i.e., water).

4. Line 218. The method of transformation of the data into an index is explained and referenced, but I struggled with the second transformation ("…relative abundance was then divided by the maximum relative abundance for a given species across all samples"). Once you do this you lose information about relative abundance of the different species within a sample, which seems like an important finding. Why not just use the %reads within a sample for the comparisons with other technologies? You need to make it more clear what is the purpose of the second transformation. I appreciate that the untransformed read data is given in Table S6, that is helpful.

5. Line 296. Please explain further how species assignment was done. Is there a lumping threshold? For example, what happened to the 1,877 ASVs from MiFish Teleost set, given that a total of 89 fish species were identified. This paragraph needs a statement about non-fish ASVs, especially as these are listed in Supplemental Tables.

6. Line 432. Following up on Comment 4, were the seine and BRUV counts also converted to indices of relative abundance across samples? If not, please explain why comparing an index to raw counts makes sense.

Other comments

Line 3. What does "Scoping the line up" mean? A google search produced links on setting up rifle scopes and examining sewer lines.

Line 46. Please clarify what "low site-species overlap between methods" means.

Line 87. The timetable for post-field sampling with eDNA seems like a theoretical minimum and leaves out requisite bioinformatic processing. In practice it is usually months between sample collection and bioinformatic processing.

Line 105. eDNA might be from "recently dissociated cells of organisms" but there are other possibilities. You could just say something like "DNA shed from organisms".

Line 124. Maybe change "students"?

Line 199. Here or in Supplemental please provide proportion of extract was used for amplification (1ul of 100ul total volume?)

Line 258. Elsewhere it says some species were excluded due to reads being below occupancy thresholds. Here it says "either a single sequence or a single individual…was treated as a detection". Please clarify.

Line 260. Can you explain this—how do you define what is are "true positive" and "true negative". I get that there is software program but it should be able to be explained in text. I don't see how you can be sure what is a true positive and what is a true negative.

Line 289. The average species/site by eDNA (34) vs seine (4), BRUV (6) is impressive, I thought you could highlight that in abstract.

Line 410. I don't understand this sentence—for one, on line 345 giant kelpfish is listed as a plus for eDNA.

Line 416. What are "sample coverage estimates"?

Line 491. Why didn't bioinformatics distinguish the surfperches if they had reference barcodes that were "nearly identical", i.e. they had differences?

Line 500. What is your "occupancy threshold of detection"? Does it make sense to screen these species out given that they were found by other methods?

Line 655. Statement about data availability doesn't seem to satisfy PLOS Data policy.

Line 893. Journal missing.

Supplemental. What does "NA" refer to in Table S5? It seems to have been detected in most samples.

Reviewer #2: The current manuscript provides a thorough comparison of three different methodologies to assess fish diversity in surf zones. Overall I found the manuscript well written and agreeable to read. The experimental design the authors used is robust. The analyses are appropriate and the results are well presented and discussed. I do not have any major concerns regarding this manuscript. I only have a couple of minor moments that I outline below.

Line 124. "previous students". I think the authors meant "previous studies".

lines 194-195. "three water replicates". It would be good to indicate explicitly here that these replicates are for each sampling site.

eDNA methods: Many publications describe and recommend the use of negative controls at different steps of the sample processing, including negative controls of the filtering process using some sort of "pure water". Here, the authors indicate they included negative controls during the PCR process only (supplementary methods). If there is a good justification for excluding the negative control during the filtering step, then it would be good to include this information in the methods. Also it would be important to indicate how many negative controls were included for PCR and in general more information about them.

6. PLOS authors have the option to publish the peer review history of their article (what does this mean?). If published, this will include your full peer review and any attached files.

Reviewer #1: No

Reviewer #2: No

---

## [Author Response · Author response to Decision Letter 0]

15 Mar 2023

Reviewers' comments:

Reviewer's Responses to Questions

Comments to the Author

1. Is the manuscript technically sound, and do the data support the conclusions?

Reviewer #1: Partly

Reviewer #2: Yes

2. Has the statistical analysis been performed appropriately and rigorously?

Reviewer #1: Yes

Reviewer #2: Yes

3. Have the authors made all data underlying the findings in their manuscript fully available?

Reviewer #1: No

Reviewer #2: Yes

4. Is the manuscript presented in an intelligible fashion and written in standard English?

Reviewer #1: Yes

Reviewer #2: Yes

5. Review Comments to the Author

Reviewer #1: This interesting report compares eDNA to two alternative technologies, hand-operated seining and baited remote underwater video (BRUV), for surveying fish presence and abundance in surf zone of sandy beaches. This is of considerable importance as the surf zone is a difficult habitat to survey by capture and visual methods. Furthermore this is one of a small number of studies that directly compare eDNA to concurrent surveys with other methods, an approach that is essential to advancing adoption of eDNA technology.

I have few major comments/suggestions, plus a number of other points that I think would improve clarity of manuscript.

Major comments

1. I suggest lumping together the seine/BRUV species that cannot be distinguished by eDNA, and then do the analysis. I think this would give a clearer picture of the relative performance of the two methods. By my analysis surfperches (Embocitidae) were 2nd most abundant by reads and 3rd most abundant by detections--it is strange to leave these out of direct comparisons with seine/BRUV, e.g., Fig. 4.

We have followed the reviewer’s advice and aggregated species to matching taxonomic ranks for both surfperches (Embocitidae) and Syngnathus sp. with identical 12S barcodes. Please see new Figure 4 and Supplemental Figures.

2. Can you find a specimen of white seabass (Atractoscion nobilis) to generate a 12S reference sequence? This is less important given that only 2 specimens were caught, but still would add depth to results.

We were able to acquire a white seabass tissue sample and successfully generated a reference barcode for this species. Upon reanalysis using the updated reference library eDNA approaches detected Atractoscion nobilis at numerous sites. 

3. Were there any negative field controls? If not please address. Also I didn't see a statement about results with negative laboratory controls (i.e., water).

We included negative controls at all stages of the eDNA work and better clarified our methods as follows:

Line 224: “Four field blanks consisting of 1 L of tap water were filtered in the field to serve as field and extraction controls.” 

Line 228: “Each unique PCR reaction included both positive and negative PCR controls; negative controls substituted molecular grade water in place of the DNA extraction, and we used either American alligator (Alligator mississippiensis) or dromedary (Camelus dromedarius), species non-native to California, for positive controls. In total, 2 positive controls, 5 PCR negative controls, and 4 field blanks were sequenced.”

Line 279: “We note that this approach does not use sequenced negative controls or field blanks to correct reads as previous work has demonstrated that this frequently removes the most abundant ASVs which arise as a result of index hopping (Kelly et al. 2018).”

Line 398: “Furthermore, six species of fish were detected in our field and negative controls when sequenced, despite the fact that no bands were visualized in PCR products during gel electrophoresis. Five of these six species were in the top 7 most abundant ASVs and in total represented 0.05% of all reads across all samples (n= 89,521). Given the prevalence of these species across all samples and technical replicates we did not use blanks for decontamination purposes (Kelly et al. 2018).”

4. Line 218. The method of transformation of the data into an index is explained and referenced, but I struggled with the second transformation ("…relative abundance was then divided by the maximum relative abundance for a given species across all samples"). Once you do this you lose information about relative abundance of the different species within a sample, which seems like an important finding. Why not just use the %reads within a sample for the comparisons with other technologies? You need to make it more clear what is the purpose of the second transformation. I appreciate that the untransformed read data is given in Table S6, that is helpful.

We agree with the reviewer that by using the eDNA index transformation we lose information on sequence reads between taxa. However, we caution that such information from sequence reads is biased. First, metabarcoding is compositional and thus can provide relative abundance information at best (Gloor et al. 2017). Second, given the strong effect of amplification bias between taxa in metabarcoding, the observed proportions are skewed (Shelton et al. 2022, Kelly et al. 2019, McLaren et al. 2019, Silverman et al. 2021). Thus, we followed the recommendations of Kelly et al. 2019 to apply the eDNA index to account for amplification biases within our eDNA metabarcoding data. By using this approach we err on the side of amplification efficiencies being a significant contributor to differences in observed sequence reads between species (Shelton et al. 2022). Accurately accounting for such biases and developing highly quantitative metabarcoding results is outside the scope of this study, although is a strong line of our future research objectives see Shelton et al. 2022. We provide all the data and code to reanalyze the results if a more appropriate transformation arises in the future.

See Lines 283-304 for in depth discussion.

5. Line 296. Please explain further how species assignment was done. Is there a lumping threshold? For example, what happened to the 1,877 ASVs from MiFish Teleost set, given that a total of 89 fish species were identified. This paragraph needs a statement about non-fish ASVs, especially as these are listed in Supplemental Tables.

We included a more detailed description of how taxonomic assignments were done in the methods section. We did not go into lengthy detail within this manuscript of the full taxonomic assignment methodology and point the reviewer to the many papers cited within this section that describe the methodology in gruesome detail (See Curd et al. 2019 appendices). We note that this manuscript builds off of extensive work conducted by the coauthors to improve taxonomic assignments of fish species in Southern California See Curd et al. 2019 and Gold et al. 2021.

Line 253-267:” Taxonomy was assigned using the Bayesian Lowest Common Ancestor classifier (Gao et al. 2017) and a curated reference database composed of fishes from the California Current Large Marine Ecosystem supplemented with the generated Atractoscion nobilis sequence following Gold et al. (2021; See detailed description in supplement). We processed each sequencing library twice using two different barcoding reference libraries. First, to assign taxonomy to marine mammalian and avian species, we used the CRUX-generated-12S database, comprised of reference barcodes for all publicly available 12S barcodes. Second, we used a curated metabarcoding database specific to California coastal marine fish to generate taxonomic assignments for fishes. We employed a Q score cutoff of 30 and Bayesian taxonomic cutoff score of 60 following the methods of Gold et al. (2021). The resulting taxonomic tables were transferred into R for further processing (R Core Team, 2020a). Reads from multiple ASVs with identical assigned taxonomic paths were summed together following the methods of Gold et al. (2021). For example, both ASV 1 and ASV 3 were assigned to Atherinops affinis with 1.5 million reads and 1.4 million reads respectively and were thus all reads assigned to both ASVs were summed.”

6. Line 432. Following up on Comment 4, were the seine and BRUV counts also converted to indices of relative abundance across samples? If not, please explain why comparing an index to raw counts makes sense.

We agree with the author that conducting comparisons of abundance between metabarcoding data and seine and BRUV counts is flawed. However, the main reason this is flawed is because of the compositional nature of metabarcoding data (Gloor et al. 2017) which results in inherently proportional abundances. Importantly, the eDNA index transformation attempts to account for species amplification biases which skew the observed compositional dataset, allowing for within species comparisons of abundance across samples (Kelly et al. 2019). In fact, we argue that given the effects of amplification efficiencies on raw observed sequences such comparisons are deeply flawed as the number of observed sequence reads for a given species is a function of not only the underlying abundance of the species but also the sequencing depth and who else is present in the sample given the competitive nature of the PCR amplification process (See Shelton et al. 2022). Thus we followed the advice of Kelly et al. 2019 which addresses this issue by employing the eDNA index transformation.

Ultimately, we recognize that our relative abundance comparisons are inherently flawed since we fail to mechanistically account for the numerous factors that influence observed sequence from biomass in the environment including fate and transport, shedding and degradation rates, and amplification efficiency biases among others (See Shelton et al. 2022). We decided to still include these supplemental analyses to highlight the difficulty of conducting such comparisons, especially given the prevalence of such comparisons in the field. We include a discussion of this issue to raise awareness of the compositional nature of metabarcoding within the broader eDNA field and the need for further research to understanding the mechanisms that relate observed sequence reads and underlying biomass of target species.

Furthermore, here we are specifically comparing an index of abundance to the total mean individuals found in replicate seine hauls and the mean MaxN statistic, the maximum number of individuals of one species in one frame during the hour-long footage. We note that the MaxN statistic is also an index of abundance since it is not relying on the total number of individuals seen, but the maximum number of individuals seen in one frame within a given period of time. Comparing an index of abundance versus absolute abundances estimates are not in and of themselves flawed as both should track the absolute change in abundance of a given species across sites. 

Line 368: “Lastly, to assess the ability of eDNA to capture relative abundance, we compared mean eDNA index scores to both the average catch counts per seine as well as average MaxN counts per BRUV station using species-specific linear regressions. Similarly, we compared BRUV-derived average MaxN counts against average seine counts. We focused our analyses on species detected jointly by each method at three or more sites. We note that comparing uncorrected compositional results (eDNA metabarcoding data) to estimates of absolute abundance (BRUV MaxN and seine counts) is inherently flawed (Kelly, Shelton & Gallego, 2019; Shelton et al., 2022). We present such results here to highlight the caveats of such approaches and discuss their merits (See Discussion).”

Line 683: “Given the observed low overlap in species detections across survey methods, assessing the capability of eDNA approaches to estimate relative abundance was challenging, particularly since eDNA surveys frequently detected a species at multiple sites where seine and BRUV surveys did not detect that species at all. This presents a core challenge of comparing eDNA to capture and visual surveys when the true abundance of species is unknown (Table 1) (Kelly et al., 2017). However, given that the ability to estimate relative abundance is a function of the ability to detect a given species, our results suggest that eDNA approaches are more sensitive and better suited than capture and visual survey methods to estimate abundance (Royle & Link, 2006; Schmidt et al., 2013; Chambert et al., 2018). This result, however, is highly dependent on the ability of eDNA approaches to resolve a given taxa. Here eDNA approaches using the MiFish-U primer set failed to resolve the most abundant surf zone species from both seine and BRUV surveys, the surfperches (Family Embiotocidae). 

Recent work from studies with greater survey overlap show promise for estimating relative abundance using eDNA approaches (Shelton et al., 2019; Di Muri et al., 2020; Stoeckle et al., 2021), particularly when accounting for the underlying mechanisms that relate observed sequence read counts to the underlying biology and biomass of detected species (Shelton et al. 2022, Yates et al. 2022). Here we note that we did not account for such underlying mechanisms in our comparisons of relative abundance which may explain why we observed such poor correlations. Such poor correlations between our our simple eDNA index transformation and seine and BRUV counts are expected given the compositional nature of metabarcoding (Gloor et al. 2017), as such a transformation does not address the many underlying non-linear factors that affect the relationship between true environmental abundance and observed reads obtained from eDNA metabarcoding sequencing (Shelton et al. 2022, McLaren et al 2019, Silverman et al. 2021). Future work should account for the suite of mechanisms that affect observed sequence reads including transport, residence time, and variation in species specific shedding and degradation rates of eDNA (Barnes & Turner, 2016) (Yates et al. 2022) (Di Muri et al., 2020). Importantly such efforts must account for the role of amplification efficiency for biasing metabarcoding results using joint models of metabarcoding data, amplification efficiency estimates, and species specific absolute abundance estimates (Kelly, Shelton & Gallego, 2019; McLaren, Willis & Callahan, 2019) (Shelton et al. 2022).”

Other comments

Line 3. What does "Scoping the line up" mean? A google search produced links on setting up rifle scopes and examining sewer lines.

We have edited the title to “A Comparison of Biomonitoring Methodologies for Surf Zone Fish Communities”.

Line 46. Please clarify what "low site-species overlap between methods" means.

We clarified this sentence as follows: Line 47 “In frequent co-detection of species between methods limited comparisons of richness and abundance estimates, highlighting the challenge of comparing biomonitoring approaches.” 

Line 87. The timetable for post-field sampling with eDNA seems like a theoretical minimum and leaves out requisite bioinformatic processing. In practice it is usually months between sample collection and bioinformatic processing.

We agree that this is a theoretical minimum for eDNA metabarcoding approaches and have clarified. To date most eDNA metabarcoding is conducted by graduate students or small labs that do not have the requisite infrastructure to analyze data quickly and at scale. However, many commercial companies and government lab facilities can provide very fast data turnaround times of sequencing data on the order of days as demonstrated by large scale covid waste water sequencing efforts. 

Line 105. eDNA might be from "recently dissociated cells of organisms" but there are other possibilities. You could just say something like "DNA shed from organisms".

We have followed the reviewer’s advice on Line 115.

Line 124. Maybe change "students"?

We have updated this sentence on Line 130 as follows: “In addition, eDNA samples are simple to collect, encouraging student led and community science, and are also cost effective, permitting increased sampling efforts across both time and space (Biggs et al., 2015; Deiner et al., 2017; Freiwald et al., 2018; Meyer et al., 2021).”

Line 199. Here or in Supplemental please provide proportion of extract was used for amplification (1ul of 100ul total volume?)

We included the following sentence on Line 17 in the supplemental methods of the elution volume: “Final elution volumes were 100 µL.”

Line 258. Elsewhere it says some species were excluded due to reads being below occupancy thresholds. Here it says "either a single sequence or a single individual…was treated as a detection". Please clarify.

We thank the reviewer for their comment. We have clarified that we implemented two separate site occupancy steps, one during the decontamination process, and another in our analyses of the cleaned or “decontaminated” data set. We have included the following sentence in Line 331: “We note this site occupancy analysis is separate from the method implemented in the decontamination process and was conducted on the final quality controlled data set.”

Line 260. Can you explain this—how do you define what is are "true positive" and "true negative". I get that there is software program but it should be able to be explained in text. I don't see how you can be sure what is a true positive and what is a true negative.

We have included the following sentence in the text to clarify:

Line 343: “We emphasize that the true occupancy of any given species at any site is unknown. Here, we are estimating the true positive (P11) and false positive (P10) of species being present at a given site using detections across repeated sampling events. Importantly, we make the explicit assumption that any detection of a species by a method is a real detection of that species.”

In our modeling approach to estimate true and false positive probability we rely on the pattern of occurrence for a given taxon within a given site across repeated sampling events. For example, a species that was detected across all repeated beach seines at every site would have a significantly higher true positive probability than a species that was only detected in 25% (1/4) beach seines at half of the surveyed sites. We recognize the terminology is cumbersome but rely on the site occupancy literature and their vernacular for our descriptions.

From the supplemental methods: “Each pattern of occurrence for a given taxon within a given site was considered a case (e.g. 2 detections out of 4 seine tows). Each unique model was run 10 times in order to filter out cases in which the model converged into a local maximum. For eDNA approaches, we employed a hierarchical model accounting for multiple technical replicates nested within bottle replicates for the eDNA surveys. In addition, for eDNA methods, we summarized the number of occurrences of each case and ran each case through a separate occupancy model to reduce computational time. 

We used the same reasonably informative priors for parameter estimations for each survey method. First, we assume that our methods do a reasonably good job of detecting species, if the species are present [13,14]. Thus, true positive probability (P11) was modeled with priors from a left-skewed beta distribution where alpha = 2 and beta = 2. Occupancy probability (Psi) modeled with using weak priors from a left-skewed beta distribution where alpha = 2 and beta = 2 assuming that most species are common across sites in this study. Lastly, we assumed that the false-positive rate of detection is unlikely to approach the true-positive rate. Thus, false positive probability (P10) was modeled with priors from a right-skewed beta distribution where alpha = 1 and beta = 10. Stan occupancy models are included in Supplemental Materials.”

Line 289. The average species/site by eDNA (34) vs seine (4), BRUV (6) is impressive, I thought you could highlight that in abstract.

We have included the following sentence in the abstract on Line 42: “On average, eDNA detected over 5 times more species than BRUVs and 8 times more species than seine surveys at a given site.”

Line 410. I don't understand this sentence—for one, on line 345 giant kelpfish is listed as a plus for eDNA.

We have rewritten this paragraph to improve clarity Line 490:” The three methods yielded different levels of detection both overall and of individual species of surf zone fish. Our eDNA samples more consistently detected 96.9% (31/32) of all species jointly observed by either BRUV or seines. Leopard shark (Triakis semifasciata) was the only species more frequently detected with BRUV (15/18) than eDNA methods (14/18). When grouping surfperch species with identical 12S reference barcodes at the family level, eDNA detected surfperches as frequently as seine surveys (17/18) (Figure 4). When grouping all Syngathus sp. pipefish species with identical 12S reference barcodes at the genus level, eDNA detected Syngathus sp. More frequently than seine and BRUV surveys.

Comparing only BRUV and seine surveys, our BRUV surveys detected elasmobranchs and flatfishes (Families Pleuronectidae and Paralichthyidae) more frequently than seine surveys. In contrast, seine surveys more frequently detected barred surfperch (Amphistichus argenteus), walleye surfperch (Hyperprosopon argenteum), California corbina (Menticirrhus undulatus), northern anchovy (Engraulis mordax), giant kelpfish (Heterostichus rostratus), and kelp pipefish (Syngnathus californiensis) than BRUV surveys.”

Line 416. What are "sample coverage estimates"?

We included the following sentence in the methods Line 323: “We estimated sample coverage, the fraction of the total incidence probabilities of the discovered species for a set of sampling units, from rarefaction and extrapolation models for species richness (Hill number q=0) for each method using the iNext package (version 2.0.20) (Hsieh, Ma & Chao, 2016).”

Line 491. Why didn't bioinformatics distinguish the surfperches if they had reference barcodes that were "nearly identical", i.e. they had differences?

The three species had multiple unique but shared reference sequences. We changed the sentence to the following line 587:” Importantly, we note all three species had shared corresponding 12S reference barcodes (Gold et al., 2021).”

Line 500. What is your "occupancy threshold of detection"? Does it make sense to screen these species out given that they were found by other methods?

Upon re running the analysis with the new reference database including the supplemental barcoding efforts these results and discussion are no longer relevant. Interestingly, school shark was detected in a technical replicates upon re analysis and we confirmed that this is stable upon multiple successive bioinformatic runs. We suspect there was a minor error in the code that has since been corrected. Likewise, upon further examination we determined that kelp pipefish shares a 12S barcode with the bay pipefish and that our pipeline could not accurately identify either of these species. We have thus adjusted our results and discussion accordingly.

Line 655. Statement about data availability doesn't seem to satisfy PLOS Data policy.

Line 893. Journal missing.

Supplemental. What does "NA" refer to in Table S5? It seems to have been detected in most samples.

This is a sequence that can not be assigned to any taxa. This species is likely an off target uncultured bacteria species that cross amplifies for the MiFish primer set Gold et al. 2021.

Reviewer #2: The current manuscript provides a thorough comparison of three different methodologies to assess fish diversity in surf zones. Overall I found the manuscript well written and agreeable to read. The experimental design the authors used is robust. The analyses are appropriate and the results are well presented and discussed. I do not have any major concerns regarding this manuscript. I only have a couple of minor moments that I outline below.

Line 124. "previous students". I think the authors meant "previous studies".

We have addressed the typo.

lines 194-195. "three water replicates". It would be good to indicate explicitly here that these replicates are for each sampling site.

We have corrected line 218:” We collected three replicate 0.5 L samples of seawater (herein referred to as sample replicates) using sterile collapsible enteral feeding bags at each site.”

eDNA methods: Many publications describe and recommend the use of negative controls at different steps of the sample processing, including negative controls of the filtering process using some sort of "pure water". Here, the authors indicate they included negative controls during the PCR process only (supplementary methods). If there is a good justification for excluding the negative control during the filtering step, then it would be good to include this information in the methods. Also it would be important to indicate how many negative controls were included for PCR and in general more information about them.

We have clarified our use of field blanks and negative controls in this study. Please see the above comment.

6. PLOS authors have the option to publish the peer review history of their article (what does this mean?). If published, this will include your full peer review and any attached files.

Do you want your identity to be public for this peer review? For information about this choice, including consent withdrawal, please see our Privacy Policy.

Reviewer #1: No

Reviewer #2: No

---

## [Decision Letter · Decision Letter 1]

26 Apr 2023

A Comparison of Biomonitoring Methodologies for Surf Zone Fish Communities

PONE-D-21-35964R1

Dear Dr. Gold

Based on the judgment of experts in the field and since we consider your study has been improved significantly from the previous version. We’re pleased to inform you that your manuscript has been judged scientifically suitable for publication and will be formally accepted for publication once it meets all outstanding technical requirements.

Kind regards,

Alejandro Pérez-Matus

Academic Editor

PLOS ONE

Additional Editor Comments (optional):

Dear Authors

On behalf of reviewers that have mentioned that authors have improved the manuscript based on constructive comments. I consider your manuscript suitable for publication. Congratulations

Reviewers' comments:

Reviewer's Responses to Questions

**Comments to the Author**

1. If the authors have adequately addressed your comments raised in a previous round of review and you feel that this manuscript is now acceptable for publication, you may indicate that here to bypass the “Comments to the Author” section, enter your conflict of interest statement in the “Confidential to Editor” section, and submit your "Accept" recommendation.

Reviewer #2: All comments have been addressed

2. Is the manuscript technically sound, and do the data support the conclusions?

Reviewer #2: Yes

3. Has the statistical analysis been performed appropriately and rigorously? 

Reviewer #2: Yes

4. Have the authors made all data underlying the findings in their manuscript fully available?

Reviewer #2: Yes

5. Is the manuscript presented in an intelligible fashion and written in standard English?

Reviewer #2: Yes

6. Review Comments to the Author

Reviewer #2: I think the authors have addressed all comments from the previous round of reviews. I have no further comments. Congrats to the authors.

7. PLOS authors have the option to publish the peer review history of their article (what does this mean?). If published, this will include your full peer review and any attached files.

Reviewer #2: No

---

## [Editor Report · Acceptance letter]

19 May 2023

PONE-D-21-35964R1 

A Comparison of Biomonitoring Methodologies for Surf Zone Fish Communities 

Dear Dr. Gold:

I'm pleased to inform you that your manuscript has been deemed suitable for publication in PLOS ONE. Congratulations! Your manuscript is now with our production department. 

Kind regards, 

on behalf of

Dr. Alejandro Pérez-Matus 

Academic Editor

PLOS ONE